# Unmet Primary Health Care Needs among Nepalese Immigrant Population in Canada

**DOI:** 10.3390/healthcare11152120

**Published:** 2023-07-25

**Authors:** Bishnu Bahadur Bajgain, Mohammad Z. I. Chowdhury, Rudra Dahal, Kalpana Thapa Bajgain, Kamala Adhikari, Nashit Chowdhury, Tanvir C. Turin

**Affiliations:** 1Department of Community Health Sciences, Cumming School of Medicine, University of Calgary, Calgary, AB T2N 4N1, Canada; 2Community Scholar and Citizen Researcher, Nepalese-Canadian Community, Calgary, AB T2N 1N4, Canada; 3Department of Family Medicine, University of Calgary, Calgary, AB T2N 4N1, Canada; 4Faculty of Health Sciences, University of Lethbridge, Lethbridge, AB T1K 1M4, Canada; 5Department of Population and Public Health, Alberta Health Services, Calgary, AB T2W 1S7, Canada

**Keywords:** unmet needs, primary healthcare, healthcare system, access, barriers, equity, Nepalese, immigrant

## Abstract

Background: Immigrants represent over one-fifth (21.9%) of the Canadian population, which is an increasing trend. Primary care is a gateway to accessing the healthcare system for the majority of Canadians seeking medical services; however, Canada reported a growing shortage of healthcare providers, mainly primary care practitioners. Canadians, including immigrants, encounter many unmet healthcare needs due to various reasons. This study aimed to assess unmet healthcare (UHC) needs and associated factors among Nepalese immigrants residing in Calgary. Methods: A cross-sectional study using a self-administered questionnaire was conducted in 2019. UHC needs were measured based on a single-item question: “During the past 12 months, was there ever a time that you felt you needed medical help, but you did not receive it”. A follow-up question was asked to learn about associated unmet needs factors, and the responses were categorized into availability, accessibility, and acceptability. Descriptive and multivariable logistic regression was employed to assess the association between UHC needs and its predictors by using STATA version 14.2. Results: Of 401 study participants, nearly half of the participants (*n* = 187; 46.63%) reported UHC needs, which was not significantly different among male and female participants (*p* = 0.718). UHC needs were nearly two times higher among those aged 26–45 (AOR 1.93) and those ≥56 years (AOR 2.17) compared to those under 25 years of age. The top reasons reported for unmet needs were long waits to access care (67.91%), healthcare costs (57.22%), and lack of knowing where to get help (31.55%). Overall, “services availability when required” was a leading obstacle that accounted for UHC needs (*n* = 137, 73.26%). Nearly two-thirds (*n* = 121, 64.71%) of participants reported that “accessibility of services” was a barrier, followed by “acceptability (*n* = 107, 57.22%). Those who reported UHC needs also reported an impact on their lives personally and economically. The most commonly reported personal impact was mental health impact, including worry, anxiety, and stress (67.38%). The most common economic impact reported due to UHC needs was increased use of over-the-counter drugs (33.16%) and increased healthcare costs (17.20%). Conclusions: UHC needs are presented in the Nepalese immigrant population. Accessibility to healthcare is limited for several reasons: waiting time, cost, distance, and unavailability of services. UHC needs impact individuals’ personal health, daily life activities, and financial capacity. Strategies to improve access to PHC for disadvantaged populations are crucial and need to be tackled effectively.

## 1. Introduction

Primary Health Care (PHC) is a fundamental component of the healthcare system in many countries. PHC addresses the major health issues of individuals and families in the community and serves as the first level of contact within the healthcare system. PHC encompasses various curative and preventive services, including diagnosis and treatment of chronic and acute conditions as well as health promotion and education [1]. PHC warrants a whole-of-society approach that stems from three foundational components—meeting people’s lifelong healthcare needs, addressing broader determinants of health, and empowering patients, families, and communities to take charge of their health [2]. Canada has a universal healthcare system, meaning it has a single healthcare system across the country. By following a fundamental principle of the Canadian Health Act [3] (i.e., universal, transferable, comprehensive, accessible, and public), each province and territory organizes healthcare services according to its context. Primary care is the gateway to accessing the healthcare system for the majority of Canadians seeking medical services.

Access to healthcare is a critical determinant of health and a dynamic process that involves the individual seeking care and the healthcare delivery process and system, which has various factors that facilitate this exchange. Equitable healthcare access across populations considering socio-economic factors, is crucial [4]. Otherwise, people with limited resources and the ability to access care may not receive what they need due to various reasons, including the healthcare delivery system, cost of services, or/and individual circumstances and behaviors. Usually, self-reported unmet healthcare (UHC) needs are a commonly used indicator to measure access to healthcare services. UHC needs are the difference between services believed necessary to deal appropriately with the health problems and services actually received, which may occur from the insufficient or/and untimely delivery of services to tackle health problems [5,6].

Immigrants represent over one-fifth (21.9%) of the Canadian population, which is an increasing trend [7]. The study shows that new immigrants reported around two and a half times more difficulties in accessing primary care than the native-born population [8]. A study examining the Canadian Community Health Survey (CCHS) data conducted in 2015–2016 found increased unmet home care needs among immigrants. Recent immigrants had the highest level of unmet home care needs (43.8%) compared to long-term immigrants (40.5%) and non-immigrants (32.7%) [9]. A study reported that around 11.2% of Canadians did not receive healthcare when they felt they needed it [10], which is higher among vulnerable populations, including immigrants (12%) [6]. A greater understanding of the problems in access to care and attempting to address the issues efficiently is challenging for policymakers without knowing why healthcare needs are not being met and how these reasons differ among the subgroups of the population. Reasons for unmet needs can be classified into three categories: availability of services (e.g., wait time, availability of services when required and in the area), accessibility (e.g., cost and transportation), and acceptability of available services (e.g., personal preference or individuals circumstance) [11]. 

Studies indicate that disparities in healthcare access between immigrants and native-born populations might arise due to various reasons, such as cultural and linguistic differences, socioeconomic environment and challenges, health system factors, and relationships between care providers and patients [12,13,14]. Despite the publicly funded healthcare services in Canada, various studies show that immigrants face substantial challenges while accessing PHC services, including culture, language, cost, schedule, communication, and the current structure of the healthcare system [12,15]. Only in 2011, over a quarter of million people moved to Canada as a permanent resident, out of which about 15% were from South Asia: India (10%), Pakistan (2%), Srilanka (1%), Bangladesh (1%), and Nepal (0.5%) [16]. It is estimated that 55% of Canada’s foreign-born population will report origins from Asia by 2031 [17].

Nepalese Canadians are one of the fastest-growing immigrant communities in Canada. Reports show that the Nepalese immigrant population has increased by 87%, from 11,450 in 2011 to 21,380 in 2016 [18,19]. Like other South Asian communities, Nepalese Canadians have unique traditions, cultures, languages, and self-identities. Nepalese Canadian immigrants are often lost in the shadow of other more prominent South Asian communities, which is also reflected in the literature, as there are only a handful of studies in Canada and globally on Nepalese immigrants. Our earlier qualitative studies show that Nepalese immigrants face barriers to accessing healthcare in Canada [20]. The noticeable challenges were waiting time, knowledge about the healthcare system, service availability, transportation, workplace, culture, language and communication, and healthcare costs [20]. However, as per our best knowledge, this is a novel study among Nepalese immigrants residing in Canada to explore their UHC needs in PHC. 

This study aimed to assess UHC needs and associated factors among Nepalese immigrants residing in Calgary, AB, Canada.

## 2. Materials and Methods

### 2.1. Study Design and Setting

A community-based participatory research (CBPR) [21], cross-sectional study using a self-administered questionnaire was conducted from January to June 2019 to assess the unmet healthcare needs and associated factors among Nepalese immigrants residing in Calgary, Alberta. Unmet health needs are related to delay or non-receipt of needed medical care, which are measured based on questions about whether there was any time during the past 12 months when respondent needed medical care but delayed or did not get it because of various reasons [10,22,23]. As per the CBPR framework, we engaged community scholars, general community members, community organizations, and various stakeholders from the Nepalese community in the entire research process alongside the academia-based researchers to identify individual, societal, and systematic issues via a collaborative approach. We applied a snowball sampling technique to conduct the survey. This coupled with the deeply invested community members helped us include hard-to-reach individuals in the study and improve the quality and relevance of our analysis and interpretation.

### 2.2. Study Population and Data Collection

The study population comprised 18-years or older Nepalese immigrants living in Calgary who were able to read and write in English and were willing to participate in the survey. Participants with an intellectual disability or any other condition that inhibited communication or their ability to participate in the study were excluded from it. The recruitment was undertaken in coordination with the Nepalese Community Society of Calgary (NCSC) and other local Nepalese communities. Potential participants were connected via email, social media channels, and word of mouth. Participants who completed the initial enrollment process for the survey were encouraged to convey the study information to additional contacts based on their personal and social networks in accordance with the snowball sampling technique that we employed. Potential participants were notified about the study’s goal and objectives by researchers via in-person or/and phone. Along with a survey informed consent form, the self-administered questionnaire was dispersed among the interested and eligible participants. Upon completion of the questionnaire, participants returned the filled form with their informed consent. A follow-up note was sent via telephone or in-person once to non-respondent participants two weeks after the survey form distribution. We disseminated 500 survey questionnaires to potential participants, of which 401 (80.2%) were completed and returned.

### 2.3. Survey Tool

The survey questionnaire was developed by reviewing current literature and having multiple discussions with a team of researchers. The survey consisted of 17 questions, which had two parts: socio-demographic and health status related questions and outcomes of interest (e.g., unmet healthcare needs, reasons for unmet healthcare needs, and their impacts.) The details of the questions are discussed below. The survey questionnaire was pilot tested and refined by the research team before being administered. 

### 2.4. Variable Measurements

*Outcome variable*: Our outcome variable was unmet health (UHC) needs. Examples include an individual being unable to access any health need such as visiting a family physician and having a diagnostic test or treatment when needed. This was measured as self-reported unmet health needs based on the following single-item question: “During the past 12 months, was there ever a time that you felt you needed medical help, but you did not receive it?” The response to the question was binary “yes/no”. If participants indicated that they had UHC needs, we compiled a list of PHC items with a binary “yes/no” response to determining which elements of PHC services reported unmet needs”.

When participants reported that they had unmet needs, to understand the reasons for unmet needs, a further question was asked, “why did not they get care, or what were the difficulties they experience in accessing care?” The reported reasons were grouped into three categories: accessibility, availability, and acceptability. In the context of UHC needs, “accessibility” refers to the ease of obtaining services, “availability” denotes the presence of sufficient and appropriate services, and “acceptability” signifies the cultural and social appropriateness of the services [24]. Each of these three categories represents an indicator variable indicating the presence or absence of their respective category and was formed by combining different reasons for unmet needs. We categorized the reasons for unmet needs as “accessibility” when the reasons were either “cost” or “transportation/distance”. The reasons for unmet needs were categorized as “availability” when the reasons were either “wait time” or “availability of services when required”. When the reasons for unmet needs were “too busy” or “didn’t know where to get help” or “felt it would be insufficient” or “decided not to seek care” or “negligence” or “a language problem” or “dislike the doctor/afraid”, it was referred to as “acceptability”.

*Explanatory variables:* To understand why certain individuals have UHC needs we collected their demographic and relevant socio-economic and health data. In the survey, we asked participants about their age, sex, marital status, family size, education, employment status, and income. Age was categorized into six groups: under 25, 26–35, 36–45, 46–55, 56–65, and over 66 years. Sex was classified as male or female. Marital status was divided into three categories: married, single, and other (including separated, divorced, or widowed). Family size was grouped into two categories: 3 or fewer members, and 4 or more members. Education was categorized as graduation and under graduation and below. Employment status was classified into three categories: employed (full-time and part-time), unemployed, and student. We also asked participants about their yearly household income, which was categorized into four groups: less than or equal to $25,000, $26,000–$50,000, $51,000–$75,000, and over $76,000. We also asked if they had a family doctor and extended health insurance coverage, and how long they had been in Canada (divided into two categories: less than or equal to 5 years, and over 5 years). Participants’ self-reported health rating was categorized into three groups: excellent, very good and good, and fair and poor. Lastly, the presence of any chronic diseases was categorized as yes or no.

*Other mediating factors:* Further, for those who reported UHC needs, to understand the impact due to UHC needs, we asked participants “did you feel any of the following impact(s) from the unmet need in your life” and listed the possible impact with open options (other). The response to the question was binary “yes/no”. The reported impact was grouped into two categories: personal impact and economic impact. Each of these categories represents an indicator variable indicating the impact of their respective category and was formed by combining different impacts of unmet needs. The impact of unmet needs was categorized as “personal” when the personal impact was either “mental health”, “overall health impact”, “problem with daily activities”, suffering in personal relation”, “unable to provide childcare”, and ”care unsatisfactory”. The impact of unmet needs was categorized as “economic” when the economic impact was either “increase cost of care”, “job lost”, “income lost”, “increased dependence”, “ordered medicine from back home”, and “increased use of over-the-counter drugs”. 

### 2.5. Statistical Analysis

Descriptive statistical analysis was performed on the data, and frequencies and percentages were calculated. Multivariable logistic regression was employed to assess the association between UHC needs and its predictors (sociodemographic and health status), and adjusted odds ratios (AOR) and 95% confidence intervals (CI) were calculated by controlling for sociodemographic characteristics and health status. The result was considered statistically significant when *p* < 0.05. All analyses were performed using STATA version 14.2. 

### 2.6. Ethics Approval

Ethics approval was obtained through the Conjoint Health Research Ethics Board at the University of Calgary (REB15-2325).

## 3. Results

### 3.1. Self-Reported Unmet Needs and Their Sociodemographic Characteristics

In total, 401 surveys were collected from Nepalese immigrants living in Calgary. Table 1 represents the overall sociodemographic characteristics of the study participants by unmet need status. The survey comprised almost an equal proportion of males (49.53% with no unmet needs, 51.34% with unmet needs) and females (50.47% with no unmet needs, 48.66% with unmet needs) participants, with no significant difference in unmet needs between the sexes (*p* > 0.05). 

A significant difference was found among the age groups of the participants with and without unmet needs (*p* < 0.05). While for the other age groups, the proportion of those with and without unmet needs was similar, a much smaller proportion of those with unmet needs were in the ≤25 age group (4.28%) compared to those without unmet needs (11.21%). Married people were also more likely to be among those with unmet needs as opposed to those without (93.58% with unmet needs, 83.64% with no unmet needs; *p* < 0.05).

More than half of the participants reported that their education level was undergraduate and below (60.09% with no unmet needs, 66.31% with unmet needs), with no statistical difference between groups (*p* > 0.05). Most of the participants were employed either full-time or part-time (80.37% with no unmet needs, 81.82% with unmet needs), with no significant difference (*p* > 0.05).

Most of the participants had a regular family doctor regardless of being with (96.79%) or without unmet needs (94.86%). A higher proportion of the participants without extended health insurance reported having unmet needs (78.50% vs. 70.05%); however, that was not statistically significant (*p* > 0.05). The yearly household income distribution showed a significant difference (*p* < 0.05), with a greater proportion of participants with unmet needs having an income between $26,000 and $50,000 (32.42%) than those without unmet needs (21.84%). There was not a significant difference in the length of stay in Canada between the two groups (*p* > 0.05).

In terms of self-reported health rating, it was observed that people with unmet needs were less likely to rate their health as excellent (4.52% vs. 13.15%), which was a significant difference (*p* < 0.05). Lastly, there was a significant difference in the presence of chronic diseases (*p* < 0.05), with participants with unmet needs having a greater proportion (44.92%) of chronic diseases than participants without unmet needs (19.16%).

### 3.2. Who Are More Likely to Have an Unmet Healthcare Need?

Table 2 shows the unadjusted and adjusted odds ratios for the UHC needs and their predictors, calculated using multivariable logistic regression. Before adjustment, participants in the 26–35, 36–45, 46–55, and 56–65 age groups had a significantly higher likelihood of having UHC needs (UOR 2.73 [95% CI: 1.15–6.47], 2.74 [95% CI: 1.16–6.48], 3.13 [95% CI: 1.18–8.30], and 6.75 [95% CI: 1.63–28.03], respectively) compared to those under 25 years of age. Married individuals had a higher likelihood of having UHC needs (UOR 2.76 [95% CI: 1.34–5.65]) compared to single ones, which was statistically significant. Participants with an education level of under graduation and below had a 1.31 (UOR [95% CI: 0.87–1.97]) times higher likelihood of having UHC needs than those with graduation degrees, however, this was statistically not significant. Though statistically not significant, those who had a family doctor had a 1.63 (UOR [95% CI: 0.59–4.51]) times greater likelihood of having UHC needs compared to those who did not have a family doctor. Furthermore, participants who had one or more chronic diseases were found to be more than three times (UOR 3.44 [95% CI: 2.20–5.38]) more likely to have UHC needs than those who did not have any chronic diseases.

After adjusting for the covariates, participants in the age brackets of 26–35 and 36–45 and elderly participants (56–65 years) had a higher likelihood of having UHC needs compared to those under 25 years of age (AOR 1.93 [95% CI: 0.53–7.01], 1.88 [95% CI: 0.48–7.34], and 2.17 [95% CI: 0.22–20.95] respectively). Married people retained a more than two-fold higher likelihood of having UHC needs (AOR 2.44 [95% CI: 0.83–7.21]) compared to single ones after adjustments. Similarly, people who reported their education as under graduation and below had a 1.36 (AOR [95% CI: 0.79–2.34]) times higher likelihood of having UHC needs than those with graduation degrees. Those who had a family doctor also had a 1.61 (AOR [95% CI: 0.50–5.26]) times greater likelihood of having UHC needs compared to those who did not have a family doctor, but remained statistically not significant after adjustments. Furthermore, participants who had one or more chronic diseases were found to be more than four times (AOR 4.52 [95% CI: 2.46–8.29]) likely to have UHC needs than those who did not have any chronic diseases. This was the only variable that remained significant after adjustment. Self-reported UHC needs did not differ significantly by sex, family size, employment status, extended health insurance coverage, household income, or length of stay in Canada.

### 3.3. Unmet Healthcare Needs by a List of Healthcare Services

Table 3 shows the list of services listed as UHC needs and their frequency and percentage. Of 187 individuals who reported UHC needs, the three most common areas were dental and related services (55.08%), accessing referral services (50.27%), and vision and related services (42.25%), whereas the least reported unmet needs areas were emergency services (0.53%) and injury treatment-related services (2.67%). Other major areas where unmet needs were reported were services related to family physicians (28.34%), diagnostic services (24.06%), and accessing medication/pharmacy-related services (8.02%). 

### 3.4. Unmet Healthcare Needs by the Distribution of Reasons

Table 4 shows the frequency and percentage contribution of each of the reasons for UHC needs. The top three common reasons given by 187 participants who reported that their healthcare needs were not met during the previous 12 months of this study were long wait times to access care (67.91%), healthcare cost (57.22%), and participants did not know where to get help (31.55%). Other major reasons indicated were care not available when required (27.81%), distance/transportation (17.65%), language problems (15.05%), and felt it would be inadequate (12.90%). The analysis also revealed that, of the 187 individuals who reported UHC needs, “services availability when required” was the leading obstacle that accounted for UHC needs (*n* = 137, 73.26%). Similarly, nearly two-thirds (*n* = 121, 64.71%) of participants who reported UHC needs highlighted that “accessibility of services” was a barrier accounted for. Acceptability (*n* = 107, 57.22 encompasses everything from personal characteristics to attitudes, which were other factors in determining UHC needs.

### 3.5. Self-Reported Impact Due to Unmet Healthcare Needs

Table 5 summarizes the self-reported impact due to unmet healthcare needs among Nepalese immigrants in Calgary. Participants who reported UHC needs (*n* = 187) also reported an impact on their lives personally and economically because their healthcare needs were not met in a timely manner. The most commonly reported personal impact was mental health impact, including worry, anxiety, and stress not only in their own lives (67.38%), but also in in the lives of their family or friends (19.25%). Other major personal impacts reported were increased pain and other symptoms (18.72%), increased health problems (18.72%), daily activities suffered (6.95%), overall health deteriorated (5.88%), and personal relations suffered (5.35%). Similarly, the most common economic impact reported, due to UHC needs, were increased use of over-the-counter drugs (33.16%) and increased healthcare costs (17.20%). Moreover, participants also reported that because their healthcare needs were not met on time, their dependency increased (5.88%), and they lost work (4.81%) and they lost income (3.23%). Of those who reported impact due to UHC needs, a good number of participants (15.05%) reported that they imported/used medicine from back home. 

## 4. Discussion

The study aimed to shed light on the UHC needs and associated factors of the Nepalese immigrants residing in Calgary, Alberta, Canada. Our findings reveal that socioeconomic characteristics are important factors in describing UHC needs. Nearly half of the participants (*n* = 187; 46.63%) reported UHC needs. This rate is significantly higher compared to national statistical data, that one-tenth of the immigrant population has UHC needs [6,22]. Our findings also aligned with other studies conducted among immigrant communities in British Columbia: one-third of the Punjabi and Chinese immigrants reported that their healthcare needs were not met [23]. Interesting differences were found in UHC needs; were higher among 26–45 years old (77.54%), married (93.58%), education under graduation (66.31%), working status (81.82%), and those who reported having at least one chronic disease condition (44.92%). The most common areas reported unmet needs were dental (55.08%), referral (50.27%), vision (42.25%), family physicians (28.34%), and diagnostic services (24.06%). Our results aligned with Statistics Canada’s report that revealed the largest proportion of reported unmet needs was for physical health problems (65.1%), mental health (11.4%), regular health checkups (9.4%), and treatments for injuries (8.8%) [10]. 

Our study revealed that the most common reasons for UHC needs were time to access care (wait time), cost of provided services and medication, lack of healthcare information (did not know where to get help when healthcare was needed), distance/transportation, unavailability of services, language barriers, and inadequate care. Our findings were supported by the report from Statistics Canada that the three most reported reasons for not meeting individuals’ healthcare needs were long wait time (33.40%), unavailable healthcare services (13.7%), and cost (11.4%) [10]. Our results also aligned with previous studies that highlighted barriers to accessing PHC services among the immigrant population in Canada [6,12,15,21,22,23]. Canada has a publicly funded healthcare system aimed at ensuring equitable access to care regardless of an individual’s age, gender, socioeconomic standing, or immigration status [3,24]; however, evidence suggests that immigrants are busy maintaining their day-to-day financial needs that impact accessing their healthcare needs [12,23,25,26]. 

Our findings support earlier studies [10,12,15,21,27] that the long waiting time and unavailability of services were major reasons for UHC needs and obstacles in accessing care among the immigrant population, which was also similar to individuals born in Canada [28]. Interestingly, our study revealed that of those who reported UHC needs, most of them (96.79%) also reported that they had a family doctor. A possible reason could be that the unmet needs arise from the difference between the expectation of certain services and approaches from the family doctor and the services and approach received (i.e., too busy, perceived negligence, limited health literacy, cultural and language barriers) [12,23,29,30]. Accordingly, another reason could be due to the below-average physician and population ratios in Canada compared to other developed countries [31].

Our findings revealed that an individual’s personal health and financial capacity were impacted due to UHC needs among the Nepalese immigrant population. The most commonly reported impact was mental health (e.g., worry, anxiety, and stress not only to individuals but also to family and friends) and physical health (e.g., suffering from daily activities, overall health deterioration, and personal relations suffered). Our findings aligned with previous studies where Canadians reported their health was negatively impacted due to delay in care, which was similar to individuals born in Canada and outside, and the impact was mostly in mental and physical health and in the elderly [32,33].

Similarly, the financial hurdle was a major impact reported due to UHC needs among the Nepalese immigrants, mostly increased healthcare costs such as increased use of over-the-counter drugs and even participants importing/using medicines from back home. Our findings aligned with the previous systematic review that reported that due to financial barriers, immigrants sometimes access medical care and supplies (e.g., medicine, dental care, eyeglasses) from their country of origin [12]. Moreover, our results showed that healthcare needs were not met in a timely manner, resulting in increased dependency, lost work, and income. Unmet healthcare needs are presented and most of the reasons are similar among immigrant populations. Our result aligns with other studies conducted among various ethnic community including Punjabi, Chinese, and Bangladesh [23,27]. 

### Strengths and Limitations of the Study

To our knowledge, this is the first study conducted to assess the UHC needs among Nepalese immigrants residing in Calgary, Alberta, Canada. This study has significant contributions to understanding UHC needs and associated factors. The strength of this study is that we used a community-engaged research approach [34], ensuring meaningful community involvement. Our team consisted of Nepalese community scholars and citizen researchers, and we also engaged with the Nepalese community organization (NCSC) during the research planning and execution phases, which allowed us to build a trust-based relationship with the Nepalese immigrant community in Calgary. Additionally, we used a self-administered questionnaire where the participants filled out the survey on their own time, often taken at home, allowing them to contemplate the questions regarding concepts of UHC needs and associated factors and answer reflectively and completely, increasing our data quality.

This study also has some constraints. This being a cross-sectional study, it lacks the ability to establish a causal relationship between the various sociodemographic factors and UHC needs that we found a correlation between. Because of the snowball sampling method, another limitation of this study would be the lack of generalizability of the study findings beyond the targeted sample. The participants of this study were limited to Nepalese immigrants residing in the Calgary area only, which may not be representative of other immigrants in Canada. However, this study provides significant insights into the other area of Canada where Nepalese immigrants are distilled, such as Edmonton, Toronto, Ontario, Vancouver, and British Columbia. This survey relies upon self-reported data, which is subjected to recall bias or bias of past experiences. The survey was conducted prior to the outbreak of the COVID-19 pandemic, and the pandemic has impacted all aspects of life including healthcare needs. Further study to understand how healthcare needs have been impacted since the onset of the pandemic would be valuable.

## 5. Conclusions

UHC needs are common in the Nepalese immigrant population. Accessibility to healthcare is limited due to a number of reasons, including waiting time, cost, distance, and unavailability of services. UHC needs result in an impact not only on individuals’ personal health (e.g., physical, mental, and social well-being) and daily life activities but also on individual financial capacity (e.g., increased healthcare cost, work/income loss). UHC needs may increase health inequalities that may result in poorer health. Future studies on UHC needs among various immigrant communities are needed further to understand accessibility, availability, and acceptability of care. Strategies to improve access to PHC for disadvantaged populations, including immigrant communities, need to be tackled for effective access to services is crucial. We believe that this study will expand the knowledge of policymakers, healthcare providers, and researchers about immigrants’ UHC needs and the unique factors associated with accessing PHC in Canada. The diverse ethnic group might have their own preferences and expectations of healthcare services which might differ from the existing healthcare system of Canada. More community-based research in diverse communities is crucial to understand their unique challenges and facilitators, preferences, and expectations. Thus, understanding and respecting diversity, capturing experiences and preferences, and identifying key barriers and enablers are crucial in reducing health inequalities and improving care access.

## Figures and Tables

**Table 1 healthcare-11-02120-t001:** Characteristics of the study population by unmet need status.

Independent Variable	Unmet Needs %		*p*-Value (Chi-Squared Test/Fisher’s Exact as Appropriate)
	No (*n* = 214)	Yes (*n* = 187)	
**Gender**			
Male	106 (49.53)	96 (51.34)	0.718
Female	108 (50.47)	91 (48.66)	
**Age**			
≤25 years	24 (11.21)	8 (4.28)	0.035
26–35 years	77 (35.98)	70 (37.43)	
36–45 years	82 (38.32)	75 (40.11)	
46–55 years	24 (11.21)	25 (13.37)	
56–65 years	4 (1.87)	9 (4.81)	
≥66 years	3 (1.40)	0 (0.00)	
**Marital Status**			
Married	179 (83.64)	175 (93.58)	0.005
Single	31 (14.49)	11 (5.88)	
Other (Separated/Divorced or Widowed)	4 (1.87)	1 (0.53)	
**Family Size**			
3 & less members	104 (49.06)	100 (53.48)	0.378
4 & more members	108 (50.94)	87 (46.52)	
**Educational Attainment**			
Graduation	85 (39.91)	63 (33.69)	0.199
Undergraduation & below	128 (60.09)	124 (66.31)	
**Employment status**			
Employed (FT + PT)	172 (80.37)	153 (81.82)	0.137
Unemployed	23 (10.75)	26 (13.90)	
Student	19 (8.88)	8 (4.28)	
**Family Doctor**			
Yes	203 (94.86)	181 (96.79)	0.338
No	11 (5.14)	6 (3.21)	
**Extended Health Insurance**			
Yes	168 (78.50)	131 (70.05)	0.053
No	46 (21.50)	56 (29.95)	
**Yearly household income**			
≤$25,000	26 (12.62)	20 (10.99)	0.009
≥$26,000–50,000	45 (21.84)	59 (32.42)	
≥$51,000–$75,000	66 (32.04)	67 (36.81)	
≥$76,000	69 (33.50)	36 (19.78)	
**Length of stay in Canada**			
≤5 years	93 (44.93)	81 (45.00)	0.989
5+ years	114 (55.07)	99 (55.00)	
**Overall self-reported health rating**			
Excellent	28 (13.15)	8 (4.52)	0.001
Very good & good	174 (81.69)	148 (83.62)	
Fair & poor	11 (5.16)	21 (11.86)	
**Chronic diseases**			
Yes	41 (19.16)	84 (44.92)	<0.001
No	173 (80.84)	103 (55.08)	

**Table 2 healthcare-11-02120-t002:** Unadjusted and adjusted odds ratio (OR) of unmet needs in relation to socio-demographic and health status.

Explanatory Variable	Unadjusted O.R.	[95% C.I.]	Adjusted O.R.	[95% C.I.]
**Sex**				
Male	Reference category		Reference category	
Female	0.93	[0.63–1.38]	0.64	[0.38–1.08]
**Age**				
≤25 years	Reference category		Reference category	
26–35 years	2.73	[1.15–6.47]	1.93	[0.53–7.01]
36–45 years	2.74	[1.16–6.48]	1.88	[0.48–7.34]
46–55 years	3.13	[1.18–8.30]	0.71	[0.15–3.25]
56–65 years	6.75	[1.63–28.03]	2.17	[0.22–20.95]
≥66 years	-	-	-	-
**Marital Status**				
Married	2.76	[1.34–5.65]	2.44	[0.83–7.21]
Single	Reference category		Reference category	
Other (Separated/Divorced or Widowed)	0.70	[0.07–7.00]	0.23	[0.01–4.02]
**Family Size**				
3 & less members	Reference category		Reference category	
4 & more members	0.84	[0.57–1.24]	0.75	[0.44–1.28]
**Educational Attainment**				
Graduation	Reference category		Reference category	
Under graduation & below	1.31	[0.87–1.97]	1.36	[0.79–2.34]
**Employment status**				
Employed (FT + PT)	0.79	[0.43–1.44]	0.85	[0.36–1.99]
Unemployed	Reference category		Reference category	
Student	0.37	[0.14–1.01]	0.64	[0.19–2.20]
**Family Doctor**				
Yes	1.63	[0.59–4.51]	1.61	[0.50–5.26]
No	Reference category		Reference category	
**Extended health Insurance**				
Yes	0.64	[0.41–1.01]	0.58	[0.30–1.14]
No	Reference category		Reference category	
**Yearly household income**				
≤$25,000	Reference category		Reference category	
≥$26,000–50,000	1.70	[0.85–3.43]	1.52	[0.61–3.80]
≥$51,000–75,000	1.32	[0.67–2.59]	0.90	[0.35–2.31]
≥$76,000	0.68	[0.33–1.38]	0.54	[0.20–1.44]
**Length of stay in Canada**				
≤5 years	Reference category		Reference category	
5+ years	0.99	[0.67–1.49]	0.95	[0.56–1.63]
**Overall self-reported health rating**				
Excellent	0.15	[0.05–0.44]	0.22	[0.06–0.83]
Very good & good	0.45	[0.21–0.95]	0.45	[0.16–1.25]
Fair & poor	Reference category		Reference category	
**Chronic diseases**				
Yes	3.44	[2.20–5.38]	4.52	[2.46–8.29]
No				

Note: O.R: Odds Ratio, FT: Full time, PT: Part-time.

**Table 3 healthcare-11-02120-t003:** The list of PHC elements that reported unmet needs (*n* = 187).

SN	List of Services Reported Unmet Needs	*n* (%)
1	Family physician (consultation/treatment)	53 (28.34)
2	Accessing referral and related services	94 (50.27)
3	Laboratory related services	45 (24.06)
4	Dental and related services	103 (55.08)
5	Vision and related services	79 (42.25)
6	Medication/pharmacy and related services	15 (8.02)
7	Basic emergency hospital/urgent care services	12 (6.42)
8	Emergency Medical Ambulance Service, 911 service	1 (0.53)
9	Child developmental and related services (immunization/growth of child)	13 (6.95)
10	Injury treatment-related service	5 (2.67)
11	Maternity care and related services (only female participant)	9 (4.81)
12	Rehabilitation and related services (therapy/social/medical/occupational)	8 (4.28)
13	Nutritional and related services	8 (4.30)
14	Mental well-being and psychosocial related services	15 (8.02)
15	Public health/health promotion-related services (screening, immunization, health information, etc.)	7 (3.74)
16	Other services (s), please specify…	1 (0.53)

Note: Respondents could report more than one reason for why they felt that care was not received. As a result, this table is not additive (they may add to more than 100%).

**Table 4 healthcare-11-02120-t004:** Reasons for unmet health needs (*n* = 187).

Variables	*n* (%)
**Accessibility**	121 (64.71)
Cost	107 (57.22)
Distance/transportation	33 (17.65)
**Availability**	137 (73.26)
Waiting time	127 (67.91)
Care not available when requested	52 (27.81)
Care not available in area	8 (4.28)
**Acceptability**	107 (57.22)
Too busy	5 (2.67)
Did not know where to get help	59 (31.55)
Felt it would be inadequate	24 (12.90)
Decided not to seek care	10 (5.35)
Negligence	4 (2.14)
Language problem	28 (15.05)
Disliked doctor/afraid	15 (8.02)
Personal/family responsibility	11 (5.88)
Accessibility + Availability	174 (93.05)
Accessibility + Acceptability	158 (84.49)
Availability + Acceptability	153 (81.82)

Note: Respondents could report more than one reason for why they felt that care was not received. As a result, this table is not additive (they may add to more than 100%).

**Table 5 healthcare-11-02120-t005:** Impact of unmet Health needs (*n* = 187).

Variables	*n* (%)
**Personal impact** Worry, anxiety, stress Worry or stress for family or friends Pain or other symptoms Overall health deteriorated condition/worse Health problem increased Unable to do childcare Personal relation suffered Problems with activities of daily living Care unsatisfactory Other	126 (67.38) 36 (19.25) 35 (18.72) 11 (5.88) 35 (18.72) 6 (3.21) 10 (5.35) 13 (6.95) 12 (6.42) 1 (0.53)
**Economic impact** Increased cost Loss of work Loss of income Increased dependence Use medicine that brought from back home Increased use of over-the-counter drugs	32 (17.20) 9 (4.81) 6 (3.23) 11 (5.88) 28 (15.05) 62 (33.16)

Note: Respondents could report more than one reason for why they felt that care was not received. As a result, this table is not additive (they may add to more than 100%).

## Data Availability

The data presented in this study are available on request from the corresponding author.

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
