# Peer review of "Unmet Primary Health Care Needs among Nepalese Immigrant Population in Canada"

_healthcare, 2023, doi:10.3390/healthcare11152120_

Round 1

Reviewer 1 Report

Peer Reviewer 1

 1.   Move this to the last section of the methods “Ethics approval was obtained through the Conjoint Health Research Ethics Board at the University of Calgary (REB15-2325)” [Line 118-120].

 2.   Write this (p=0.718) as (ρ>0.05) [Line 195]

 3.   Write this (p=0.035) as (ρ<0.05) [Line 197]

 4.   Kindly separate the results interpretation of the unadjusted odds ratio (UOR) from the adjusted odds ratio (AOR) with paragraphs. This will help your readers to understand the interpretation. Also, signify the significant variables in your interpretations.

 5.   Tell us what is unmet health needs in the methods? Unmet health needs can be explained or related as delay or non-receipt of needed medical care. These are based on questions about whether there was any time during the past 12 months when the respondent needed medical care but delayed or did not get it because of the cost.

 6.   The sub-section “Variables” and “measurements” should be merged. First, we should have a sub-headline as ‘Variable Measurements’, then under it we have a sub sub-section title as Outcome Variable and the next sub-section is Explanatory variables. For instance,

2.4 Variable Measurements

2.5 Outcome Variable (or Dependent variable)

2.6 Explanatory variables (or Independent variable) [Line 143-179]

7.   The outcome variable should be clearly stated and the questions that forms the outcome variable should be included in the explanation.

8.   The explanatory variables should clearly be stated and whether they will be categorized or not should be explained in detail. For instance, education was categorised as No schooling, schooling.

 9.   Change Reference to Reference Category ( or RC) in Table 2 [Page 6-7]

 10.   Can you explain in detail how you measure availability, accessibility and acceptability for unmet health needs. Kindly check acceptability …I do not think it is suitable for unmet health needs, it is applicable to fertility. I suggest you use the word ‘Satisfactory’ for acceptability.

 11.   How do you measure the economic impact and personal impact? These were not measured in the methods. Kindly insert their measurements in a separate session under variable measurements. You can tag them as other mediating factors.

 12.   Align your discussion with the objectives and findings of this study

 13.   The survey tool (questionnaire) used in this study to gather data should be vividly and explicitly discussed [Line 140-142].

 14.   The limitations of this study should be centred around the literature, methods and findings [337-359].

Paper should be sent to professional editors for editing

Author Response

Comments from Reviewer 1

  1. Move this to the last section of the methods “Ethics approval was obtained through the Conjoint Health Research Ethics Board at the University of Calgary (REB15-2325)” [Line 118-120].

Reply:

Thanks for the comment. We have moved this part to the last section of methods as suggested.

  1. Write this (p=0.718) as (ρ>0.05) [Line 225]

Reply:

We appreciate your suggestion and have now replaced (p=0.718) with (ρ>0.05) [Line 196].

  1. Write this (p=0.035) as (ρ<0.05) [Line 197]

Reply:

Thank you for your advice. The notation (p=0.035) has been updated to (ρ<0.05) [Line 227].

  1. Kindly separate the results interpretation of the unadjusted odds ratio (UOR) from the adjusted odds ratio (AOR) with paragraphs. This will help your readers to understand the interpretation. Also, signify the significant variables in your interpretations.

Reply:

Following your suggestion, we have separated the interpretations of the UOR and the AOR into distinct paragraphs for clearer understanding and have highlighted the significant variables. [Lines 252-284]

“Who are more likely to have an unmet healthcare need?

Table 2 shows the unadjusted and adjusted odds ratios for the UHC needs and their predictors, calculated using multivariable logistic regression. Before adjustment, participants in the 26–35, 36–45, 46–55, and 56–65 age groups had a significantly higher likelihood of having UHC needs (UOR 2.73 [95% CI: 1.15-6.47], 2.74 [95% CI: 1.16-6.48], 3.13 [95% CI: 1.18-8.30], and 6.75 [95% CI: 1.63-28.03], respectively) compared to those under 25 years of age. Married individuals had a higher likelihood of having UHC needs (UOR 2.76 [95% CI: 1.34-5.65]) compared to single ones, which was statistically significant. Participants with an education level of under graduation and below had a 1.31 (UOR [95% CI: 0.87-1.97]) times higher likelihood of having UHC needs than those with graduation degrees, however, this was statistically not significant. Though statistically not significant, those who had a family doctor had a 1.63 (UOR [95% CI: 0.59-4.51]) times greater likelihood of having UHC needs compared to those who did not have a family doctor. Furthermore, participants who had one or more chronic diseases were found to be more than three times (UOR 3.44 [95% CI: 2.20-5.38]) likely to have UHC needs than those who did not have any chronic diseases.

After adjusting for the covariates, participants in the age brackets of 26–35 and 36–45 and elderly participants (56-65 years) had higher likelihood of having UHC needs compared to those under 25 years of age (AOR 1.93 [95% CI: 0.53-7.01], 1.88 [95% CI: 0.48-7.34], and 2.17 [95% CI: 0.22-20.95] respectively). Married people retained a more than two-fold higher likelihood of having UHC needs (AOR 2.44 [95% CI: 0.83-7.21]) compared to single ones after adjustments. Similarly, people who reported their education as under graduation and below had a 1.36 (AOR [95% CI: 0.79-2.34]) times higher likelihood of having UHC needs than those with graduation degrees. Those who had a family doctor also had a 1.61 (AOR [95% CI: 0.50-5.26]) times greater likelihood of having UHC needs compared to those who did not have a family doctor, but remained statistically not significant after adjustments. Furthermore, participants who had one or more chronic diseases were found to be more than four times (AOR 4.52 [95% CI: 2.46-8.29]) likely to have UHC needs than those who did not have any chronic diseases. This was the only variable that remained significant after adjustment. Self-reported UHC needs did not differ significantly by sex, family size, employment status, extended health insurance coverage, household income, or length of stay in Canada.”

  1. Tell us what is unmet health needs in the methods? Unmet health needs can be explained or related as delayor non-receipt of needed medical care. These are based on questions about whether there was any time during the past 12 months when the respondent needed medical care but delayed or did not get it because of the cost.

Reply:

Thank you for guiding us on this.

We have expanded the "unmet health needs" concept in the methods section, including an explanation that pertains to delays or non-receipt of necessary medical care due to cost [Lines 116-119 and lines 152-160].

  1. The sub-section “Variables” and “measurements” should be merged. First, we should have a sub-headline as ‘Variable Measurements’, then under it we have a sub sub-section title as Outcome Variable and the next sub-section is Explanatory variables. For instance,

2.4 Variable Measurements

2.5 Outcome Variable (or Dependent variable)

2.6 Explanatory variables (or Independent variable) [Line 143-179]

Reply:

Thank you for such constructive comment. We have reorganized the sections now to reflect your suggestion. [Lines 151-106]

  1. The outcome variable should be clearly stated and the questions that forms the outcome variable should be included in the explanation.

Reply:

We appreciate your feedback. The outcome variable has now been clearly stated, and we have included the questions that form the basis of the outcome variable in the explanation. [Line 152-179]

  1. The explanatory variables should clearly be stated and whether they will be categorized or not should be explained in detail. For instance, education was categorized as No schooling, schooling.

Reply:

Thank you for your suggestion. We have clarified the explanatory variables and explained the categorization concisely. [Lines 177-192]

  1. Change Reference to Reference Category ( or RC) in Table 2 [Page 6-7]

Reply:

Based on your recommendation, we have changed "Reference" to "Reference Category” in Table 2 [Page 7-8]. Thanks for suggesting this.

  1. Can you explain in detail how you measure availability, accessibility and acceptability for unmet health needs. Kindly check acceptability …I do not think it is suitable for unmet health needs, it is applicable to fertility. I suggest you use the word ‘Satisfactory’ for acceptability.

Reply:

We are grateful for your input. We have provided explanation for those terms, and we believe that now provides sufficient rationale to use the word acceptability.

Please see lines 154-167

“In the context of UHC needs, "accessibility" refers to the ease of obtaining services, "availability" denotes the presence of sufficient and appropriate services, and "acceptability" signifies the cultural and social appropriateness of the services. Each of these three categories represents an indicator variable indicating the presence or absence of their respective category and was formed by combining different reasons for unmet needs. We categorized the reasons for unmet needs as "accessibility" when the reasons were either "cost" or "transportation/distance." The reasons for unmet needs were categorized as "availability" when the reasons were either "wait time" or "availability of services when required." When the reasons for unmet needs were "too busy" or "didn't know where to get help" or "felt it would be insufficient" or "decided not to seek care" or "negligence" or "a language problem" or "dislike the doctor/afraid," it was referred to as "acceptability" (Table 4).”

As these terms and their definitions were taken from previous well-known studies (an example citation provided below), we opted on continuing to use this term. We hope for the reviewer’s kind understanding.

Citation 25: Nelson, C.H.; Park, J. The Nature and Correlates of Unmet Health Care Needs in Ontario, Canada. Soc. Sci. Med. 2006, 62, 2291-2300.

  1. How do you measure the economic impact and personal impact? These were not measured in the methods. Kindly insert their measurements in a separate session under variable measurements. You can tag them as other mediating factors.

Reply:

Thank you for your advice. We have included the measurement methods for economic impact and personal impact under 'Variable Measurements' as ‘other mediating factors’ [Line 195-207].

  1. Align your discussion with the objectives and findings of this study

Reply:

We have revised the discussion section to better align with the objectives and findings of the study, as you suggested. [Lines 378-387]

  1. The survey tool (questionnaire) used in this study to gather data should be vividly and explicitly discussed [Line 140-142].

Reply:

In response to your feedback, we have explicitly discussed the survey tool (questionnaire) used for data collection in the methods section [Lines 146-151]. We appreciate your guidance on this matter.

  1. The limitations of this study should be centred around the literature, methods and findings [337-359].

Reply:

 We've taken your suggestion into account and have revised the limitations section to better reflect the study's literature, methods, and findings. [Lines 401-413]

Reviewer 2 Report

The study provides valuable insights into the healthcare needs of a specific immigrant population in Canada, specifically focusing on Nepalese immigrants in Calgary. It contributes to the existing literature on healthcare access and utilization among immigrants, particularly from Nepal. The research addresses a relevant and timely issue, considering that immigrants comprise a significant portion of the Canadian population, and understanding their healthcare needs is crucial for providing effective healthcare services.

Methods: The mention of a "community-based, participatory research" approach is intriguing, as it suggests the involvement of stakeholders beyond the study participants in the data collection process. However, it is not clear whether the data collected through this participatory approach was incorporated into the analysis alongside the data from the respondents. To enhance clarity, it would be beneficial if the authors provided a conceptual framework explaining the use of this approach and how it influenced the data analysis.

Discussion: It would be valuable if the authors discussed the differences in unmet healthcare needs between the Nepalese population and other ethnic migrant groups. Additionally, exploring any existing government policies or initiatives aimed at addressing these issues would be beneficial for a comprehensive discussion. Since the data collection was conducted prior to the outbreak of COVID-19, it is understood that the unmet needs were not influenced by the pandemic. However, it would be insightful if the authors discussed the potential changes or tendencies in unmet healthcare needs during and after the pandemic in the Discussion section.

Author Response

Comments from Reviewer 2

  1. The study provides valuable insights into the healthcare needs of a specific immigrant population in Canada, specifically focusing on Nepalese immigrants in Calgary. It contributes to the existing literature on healthcare access and utilization among immigrants, particularly from Nepal. The research addresses a relevant and timely issue, considering that immigrants comprise a significant portion of the Canadian population, and understanding their healthcare needs is crucial for providing effective healthcare services.

Reply:

Thank you for the appreciative feedback.

  1. Methods: The mention of a "community-based, participatory research" approach is intriguing, as it suggests the involvement of stakeholders beyond the study participants in the data collection process. However, it is not clear whether the data collected through this participatory approach was incorporated into the analysis alongside the data from the respondents. To enhance clarity, it would be beneficial if the authors provided a conceptual framework explaining the use of this approach and how it influenced the data analysis.

Reply:

Thank you for this excellent suggestion.

We have included this description of CPBR model and the influence in our analysis in the method section as below [Lines 119-126].

“As per CBPR framework, we engaged community scholars, general community members, community organizations, and various stakeholders from the Nepalese community in the entire research process alongside the academia-based researchers to identify individual, societal, and systematic issues via a collaborative approach. We applied a snowball sampling technique to conduct the survey. This coupled with the deeply invested community members helped us include hard-to-reach individuals in the study and improve the quality and relevance of our analysis and interpretation.”

  1. Discussion: It would be valuable if the authors discussed the differences in unmet healthcare needs between the Nepalese population and other ethnic migrant groups. Additionally, exploring any existing government policies or initiatives aimed at addressing these issues would be beneficial for a comprehensive discussion. Since the data collection was conducted prior to the outbreak of COVID-19, it is understood that the unmet needs were not influenced by the pandemic. However, it would be insightful if the authors discussed the potential changes or tendencies in unmet healthcare needs during and after the pandemic in the Discussion section.

Reply:

Thank you for the suggestions. We have revised the manuscript to reflect your valuable suggestions.

Discussion/Lines 383-387

“Unmet healthcare needs are presented and most of the reasons are similar among immigrant population. Our result aligns with other studies conducted among various ethnic community including Punjabi, Chinese, Bangladesh [23,28].”         

Discussion/Line 409-414 (limitation section)

“The survey was conducted prior to the outbreak of COVID-19 pandemic, and the pandemic has impacted all aspects of life including healthcare needs. Further study to understand how the healthcare needs have impacted since the onset of the pandemic would be valuable.”
